# RECOGNIZING ACTIONS USING OBJECT STATES

**Nirat Saini, Bo He, Gaurav Shrivastava, Sai Saketh Rambhatla, Abhinav Shrivastava**
University of Maryland, College Park
`{nirat,bohe,gauravsh,rssaketh,abhinav2}@umd.edu`

## ABSTRACT

Object-centric actions cause changes in object states, including their visual appearance and their immediate context. We propose a computational framework that uses only two object states, *start* and *end*, and learns to recognize the underlying actions. Our approach has two modules that learn subtle changes induced by the action and suppress spurious correlations. We demonstrate that only two object states are sufficient to recognize object-centric actions. Our framework performs better than approaches that use multiple frames and a relatively large model. Moreover, our method generalizes to unseen objects and unseen video datasets.

## 1 INTRODUCTION

Objects in real world exists in different color, shape and structure. For instance, an apple can be red or green, cut or peeled, raw or riped. Growing interest in object state transformations is still limited to image based datasets such as Isola et al. (2015) and Yu & Grauman (2014), which study a set of object-state transformations for edible objects and shoes. However, using objects states for videos is still uncommon. In Fathi & Rehg (2013), actions are detected based on changes in object states. Similar to Fathi & Rehg (2013), we hypothesize that most object-centric actions induce specific changes in object states, where state includes object's appearance and immediate context. Humans are adept at identifying such actions simply based on the state transformations. In the commonsense reasoning literature, this notion of state transformation is also known as *fluent* of an object Liu et al. (2017); Fire & Zhu (2015); Mueller (2006), defined as *the status of an object which varies over time.* For instance, a lemon's *fluent* changes from whole→pieces over time, associated with *cut* action.

In this work, we propose a computational model that embodies the intuition of leveraging object state transformations, or *fluent* of an object, to reveal the underlying action being performed. In fact, we limit our focus to the question: Can we infer actions just from emphtwo distinct object states – *start* and *end*? Note that we do not claim that more intermediate states will not be beneficial for modeling transformations (*e.g*, see Wang et al. (2016b)). However, for object-centric actions, we show that only using two states can outperform standard approaches using many more frames. Moreover, prior works in action recognition domain Zhou et al. (2018a); Simonyan & Zisserman (2014); Yang et al. (2020a) use training and test splits based on actions, *i.e*, for label *cut apple*, few videos are part of train and some are testing videos. However, we focus to learn the action or transformation of any object. Hence, we create dataset splits where for an action, objects are split up as seen and unseen objects as training and test set respectively.

Firstly, our proposed method selects two bounding boxes, representing *start* and *end* states, and then extracts their representations. Our manipulation module uses these representations and learns a *fluent* representation used to classify actions. We use cooking video datasets: EpicKitchen55 Damen et al. (2018), YouCook2 Zhou et al. (2018b), and EpicKitchens100 Damen et al. (2020), and firstly train on just EpicKitchen55 and show that our model can generalize to YouCook2 and EpicKitches100. Second, we hold out several unseen objects for all actions while training and generalize to transformations of these unseen objects in test set. We show that our approach is able to improve performance for both *object-manipulating* and *context-manipulating* actions. To summarize, our contributions are as follows: (a) developing a manipulation module with object and context modules, that can understand object state changes, (b) demonstrating that amplifying relevant information from just two states is enough to outperform architectures that use several frames, and (c) demonstrating the generalization of our approach to unseen objects datasets.

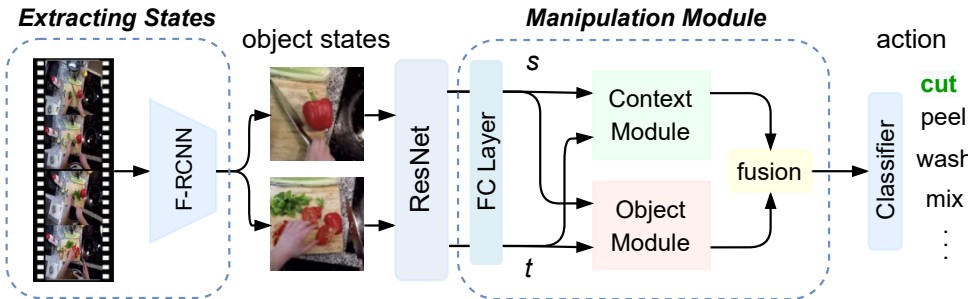

Figure 1: Our framework extracts *start* and *end* object states from the video. We use ResNet50 He et al. (2016) pre-trained on ImageNet Deng et al. (2009) as our backbone network. We introduce a novel framework to model object-transforming and context-transforming actions separately. The context module learns the subtle changes and similarities in the object environment, *i.e*, context. The object module focuses on the object states, rather than the context.

## 2 APPROACH

In this work, our goal is to recognize object-centric actions using *start→end* object state changes. A state manipulation can correspond to *object-manipulating actions* that change object's visual appearance and *context-manipulating actions* that change its context. Our approach is inspired by attention correlation between labels and images Xu et al. (2021). Our Manipulation Module shown in Figure 1 are designed to amplify similarity between these states to suppress spurious signals and amplify the differences to leverage subtle changes between *start* and *end* states.

Currently, no video dataset exists with specific object states before and after an action. Hence, as a pre-processing step, we first extract object states, or regions of the object in two distinct frames before and after the action has been performed. More details of are in provided in the appendix A. Once two distinct object states are captured, we propose our method using Manipulation Module, as shown in Figure 1. Hence, the input to our model are the *start* ($\mathcal{I}_s$) and *end* ($\mathcal{I}_t$) states. We use a pre-trained deep neural network to extract features $f_s$ and $f_t$ ($f_s, f_t \in \mathbb{R}^n$; $n = 2048$) from $\mathcal{I}_s$ and $\mathcal{I}_t$, respectively. We then pass $f_s$ and $f_t$ through an FC layer with ReLU non-linearity to obtain $g_s$ and $g_t$ ($g_s, g_t \in \mathbb{R}^n$). The features $g_s$ and $g_t$ are then passed through the Manipulation Module for further processing.

Our hypothesis is that the similarities and differences between deep features $g_s$ and $g_t$ capture the manipulation in objects and its context. Hence, our manipulation module consists of –– *Context Module* and *Object Module*. The object module captures and difference in object states, whereas context module learns to amplify the subtle similarity between object's context or environment that can be used for classification. Both module take $g_s$ and $g_t$ as input to produce representations $f_c$ and $f_o$ respectively. The two features $f_c$ and $f_o$ are then fused to produce the final representation. We describe one of the modules (the context module) in detail below and the other module is implemented in a similar fashion with minor changes.

Context Module captures the similarity between the *start* and *end* states. We first take the outer product of $g_s$ and $g_t$ denoted as $O = g_s \otimes g_t$ ($O \in \mathbb{R}^{n \times n}$). Element $o_{i,j}$ represents the similarity between $i^{\text{th}}$ element of $g_s$ with $j^{\text{th}}$ element of $g_t$. Moreover, let $\mathbf{o_{i*}}$ and $\mathbf{o_{*j}}$ represent the $i^{\text{th}}$ row and $j^{\text{th}}$ column of $O$ respectively. Then, $\mathbf{o_{i*}}$ captures the similarity of all the elements in $g_t$ with respect to $i^{\text{th}}$ element of $g_s$. To know the most similar element among $g_t$ with respect to $i^{\text{th}}$ element of $g_s$, we can take a row-wise softmax over $O$. Similarly, for $j^{\text{th}}$ element of $g_t$, column $o_{*j}$ represents the similarity with all the elements of $g_s$. Using a column-wise softmax, we can interpret the most similar and least similar element of $g_s$ with respect to $j^{\text{th}}$ element of $g_t$. Therefore, by applying column-wise and row-wise softmax, we get two matrices, $\mathcal{A}$ and $\mathcal{A}'$ ($\mathcal{A}, \mathcal{A}' \in \mathbb{R}^{n \times n}, n = 2048$).

$$\mathcal{A}[i,:] = \frac{e^{\gamma \mathbf{o_{i*}}}}{\sum_{j=1}^{d} e^{\gamma o_{ij}}} \quad \text{and} \quad \mathcal{A}'[:,j] = \frac{e^{\gamma \mathbf{o_{*j}}}}{\sum_{i=1}^{d} e^{\gamma o_{ij}}},$$

Table 1: **Quantitative results on Unseen Objects**. This table summarizes results for actions on objects not seen during training. (test set). All values are shown as mean with confidence interval (95%) over 5 splits. In the table we highlight the best and second best result in bold+underline and underline respectively.

| Network Input | Method | All Actions | | Object Transforming Actions | | Context Transforming Actions | |
|---|---|---|---|---|---|---|---|
| | | Acc. | Mean Acc. | Acc. | Mean Acc. | Acc. | Mean Acc. |
| 8 Frames | TRN Zhou et al. (2018a) | $31.49 \pm 2.2$ | $7.16 \pm 1.5$ | $18.89 \pm 6.0$ | $7.24 \pm 1.6$ | $44.87 \pm 7.1$ | $7.07 \pm 1.6$ |
| | TPN Yang et al. (2020a) | $35.60 \pm 4.1$ | $8.78 \pm 1.3$ | $\underline{28.61 \pm 9.4}$ | $8.59 \pm 2.1$ | $41.36 \pm 8.7$ | $7.42 \pm 2.5$ |
| 3 Frames | TRN Zhou et al. (2018a) | $29.78 \pm 2.5$ | $7.38 \pm 2.8$ | $16.26 \pm 6.7$ | $6.26 \pm 2.1$ | $44.16 \pm 7.7$ | $8.50 \pm 4.0$ |
| | TPN Yang et al. (2020a) | $30.59 \pm 0.7$ | $7.16 \pm 1.5$ | $18.83 \pm 9.2$ | $7.44 \pm 2.4$ | $41.14 \pm 12.2$ | $6.86 \pm 1.5$ |
| 8 Objects | TRN Zhou et al. (2018a) | $28.75 \pm 2.5$ | $7.53 \pm 1.6$ | $15.34 \pm 5.9$ | $6.35 \pm 1.7$ | $42.91 \pm 7.0$ | $8.71 \pm 2.3$ |
| | TPN Yang et al. (2020a) | $34.92 \pm 2.6$ | $9.22 \pm 2.7$ | $27.65 \pm 6.8$ | $8.51 \pm 2.3$ | $41.98 \pm 5.6$ | $7.93 \pm 3.3$ |
| 3 Objects | TRN Zhou et al. (2018a) | $27.86 \pm 3.2$ | $7.83 \pm 2.5$ | $16.70 \pm 4.7$ | $7.35 \pm 1.6$ | $39.68 \pm 7.8$ | $8.31 \pm 3.5$ |
| | TPN Yang et al. (2020a) | $28.84 \pm 3.8$ | $6.58 \pm 2.5$ | $12.30 \pm 9.8$ | $5.65 \pm 4.0$ | $\underline{45.95 \pm 14.9}$ | $7.50 \pm 1.6$ |
| 2 Objects | Fathi et al. Fathi & Rehg (2013) | $30.59 \pm 1.1$ | $8.17 \pm 1.7$ | $26.97 \pm 2.6$ | $\underline{8.81 \pm 1.1}$ | $34.00 \pm 3.5$ | $7.53 \pm 2.3$ |
| | MaxpoolResFeats | $28.37 \pm 4.1$ | $9.70 \pm 1.7$ | $18.88 \pm 2.1$ | $8.56 \pm 1.8$ | $38.47 \pm 9.3$ | $\underline{9.19 \pm 2.9}$ |
| | ConcatResFeats | $\underline{35.71 \pm 0.7}$ | $\underline{9.83 \pm 1.3}$ | $27.30 \pm 4.5$ | $7.40 \pm 2.7$ | $42.56 \pm 6.6$ | $8.37 \pm 1.0$ |
| | **Ours** | $\mathbf{38.83 \pm 1.4}$ | $\mathbf{10.08 \pm 1.6}$ | $\mathbf{29.29 \pm 8.0}$ | $\mathbf{8.96 \pm 2.9}$ | $\mathbf{47.70 \pm 7.7}$ | $\mathbf{10.79 \pm 1.4}$ |

where $\gamma$ is the inverse temperature parameter. We compute row and column sum for $\mathcal{A}$ and $\mathcal{A}'$ respectively, to get raw state masks, $\mathbf{m^t}$ and $\mathbf{m^s}$ in Eq.1 To get the final masks, we normalize with ReLU (Eq.2). We compute the final representation $f_c$ of the context module in Eq. 3, where $\odot$ is the elementwise multiplication and $W \in \mathbb{R}^{2n \times n}$ and $b \in \mathbb{R}^n$.

$$\hat{m}_j^t = \sum_{i=1}^{d} \mathcal{A}_{ij} \quad \text{and} \quad \hat{m}_i^s = \sum_{j=1}^{d} \mathcal{A}'_{ij}. \tag{1}$$

$$m^s = \text{ReLU}(\hat{m}^s - \mathbb{E}[\hat{m}^s]) \quad \text{and} \quad m^t = \text{ReLU}(\hat{m}^t - \mathbb{E}[\hat{m}^t]) \tag{2}$$

$$f_c = W^T([m_s \odot g_s; m_t \odot g_t]) + b \tag{3}$$

Similarly, we can get the object representation $f_o$, except instead of using similarity between states ($\mathcal{A}$ and $\mathcal{A}'$), use distance measure ($\mathcal{D}$ and $\mathcal{D}'$) computed as

$$\mathcal{D} = 1./\mathcal{A} \quad \text{and} \quad \mathcal{D}' = 1./\mathcal{A}'$$

where $1./*$ is the element-wise inversion of matrices. The final representation is the output of max-pooling of $f_c$ and $f_o$. We train using a linear classifier in an end-to-end fashion using softmax Cross-Entropy loss function.

## 3 EXPERIMENTS AND RESULTS

### 3.1 DATASET AND EVALUATION

We use Epic-kitchens-55 (EK-55) Damen et al. (2018) which has kitchen activities captured with an egocentric camera. Among all actions, only 13 actions are object-transforming and 30 actions are context-transforming. In total, we use 7342 videos, 2120 for object-transforming and 5222 for context-transforming actions. We create five train/val/test splits of the dataset and report the mean performance across splits. Each split has all 13 object-transforming actions and randomly chosen 13 context-transforming actions. Note that the test set, referred to as 'unseen object-actions' in each split contains actions involving objects unseen during training. We report overall accuracy and Mean accuracy. More details are provided in appendix A.

### 3.2 QUANTITATIVE RESULTS

We compare our model with two types of action recognition models:
**Spation-temporal Baselines.** Our setup is RGB based, hence we compare with the best RGB-only model for action recognition Yang et al. (2020a); Zhou et al. (2018a). TPN Yang et al. (2020a)

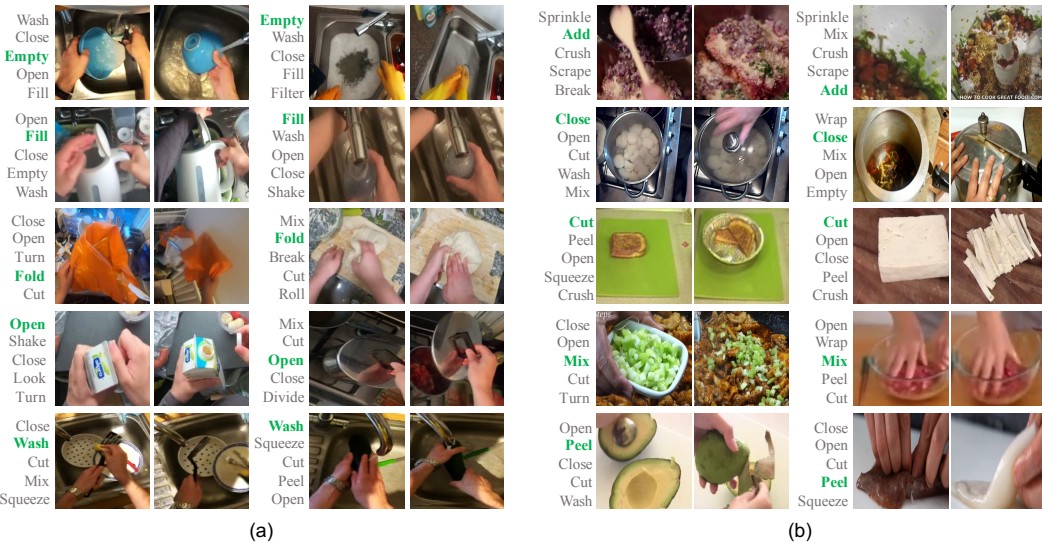

Figure 2: We show qualitative results for (a) Epic-Kitchen 100 Damen et al. (2020) and (b) Youcook2. We show top 5 predictions for object manipulating actions: empty, fill, open, peel, cut, close, and context manipulating actions: fold, wash, mix, add.

shows state-of-the-art results on multiple datasets, including EK-55. TPN Yang et al. (2020a) captures visual temporal scales of different actions. TRN Zhou et al. (2018a) accounts for relationship between frames at multiple scale (2,3,4 and 8). Both models use minimum of 3 frames, and finally consolidate the relationships between all frames. We use 3 and 8 frames for these models, with TSN Wang et al. (2016a) based sampling frames. We also train these baselines for with 3 and 8 object states, to compare with our 2 objects states setup.

**Object States Baselines.** We compare with three other baselines which use exactly as input as ours, *i.e*, 2 object states. Fathi et al. Fathi & Rehg (2013). detect changing regions in video and train SVMs for each action. We use our implementation for Fathi & Rehg (2013), with deep features for regions to replicate the results. We propose two more object states baselines: MaxpoolResFeats uses $\mathrm{maxpool}(f_s, f_t)$ and ConcatResFeats uses $\mathrm{concatenation}(f_s, f_t)$ for action classification. Note that we follow the notation explained in section 2, where $f_s$ and $f_t$ are ResNet features for *start* and *end* states. For a fair comparison, these object states baseline models have around the same number of trainable parameters as our model.

To keep it consistent with TPN Yang et al. (2020a) and TRN Zhou et al. (2018a), we use ResNet50 He et al. (2016) as our backbone network and all setups follow three-crop testing evaluation protocol. The results are reported in Table 1 for unseen object actions (test set). For 5 different splits for action classes, we denote the final accuracy as mean and confidence interval with 95% confidence, for object-transforming and context-transforming actions separately as well. Our model outperforms all the baselines for 3 and 8 frames and bounding boxes setup, not only on majority classes but also performs better for minority classes (Table 1). We show the of our object and context modules for capturing the similarity and dissimilarity, to significantly improve object-transforming actions, as compared to using max of ResNet features (MaxpoolResFeats baseline) and concatenation baseline.

### 3.3 ABLATION STUDIES

We also do an ablation study on one of the validation set split (Seen object actions), to justify our architectural choices Table 2.

**Object Baselines:** Using only ResNet features for action recognition, either maxpool (MaxpoolResFeats in Table 2) or concatenation (ConcatResFeats) of object states, also performs better than 3 and 8 frames models.

**(start→end) Attention:** This is akin to using a single module between *start* and *end* states. We use a non-spatial attention block, inspired by Wang et al. (2018); He et al. (2020). More details on how

Table 2: **Ablation analysis on Seen Objects**. We show how our amplifiers and separate modeling of object-transforming and context-transforming actions help in object-centric action recognition. All the experiments are done on split1. In the table we highlight the best and second best result in bold+underline and underline respectively.

| | All Actions | | Object Transforming | | Context Transforming | |
|---|---|---|---|---|---|---|
| | Acc. | Mean Acc. | Acc. | Mean Acc. | Acc. | Mean Acc. |
| ConcatResFeats | 65.9 | 23.23 | 72.4 | 27.56 | 48.89 | 18.49 |
| MaxPoolResFeats | 51.86 | 22.75 | 56.80 | 30.78 | 42.39 | 18.12 |
| (*start→end*) Attention Wang et al. (2018) | 58.17 | 20.95 | 65.60 | 23.69 | 39.39 | 16.22 |
| Module Alternate | 60.74 | 24.06 | 64.01 | 29.62 | 48.52 | 18.12 |
| Our Approach | **68.77** | **25.39** | **76.40** | **33.21** | **49.49** | **18.96** |

this attention is different than our method are in appendix. This setup shows we need different ways to amplify context and object features between *start* and *end* states.

**Module Alternative:** For understanding the significance of our proposed Manipulation Module, we model object manipulating actions as difference between features for object states and mean of features for context transforming actions. Fusion of the two is done in the same way with maxpool. This experiment shows need for two separate object and context modules.

### 3.4 QUALITATIVE RESULTS

We show qualitatively how our method performs on out-of-domain data, such as Epic Kitchens-100 Damen et al. (2020) and YouCook2 Zhou et al. (2018b). Although, our state detector only finds a limited set of seen objects for these datasets, we achieve, 65.60% Top1 accuracy and 90.10% Top5 accuracy for Epic Kitchens-100, for unseen and seen actions. Some examples of correctly classified actions are shown in Figure 2. Note that these samples are classified correctly by our model among top 5 predictions.

## 4 CONCLUSION

We presented a framework for recognizing object-centric actions using object states. We show that by leveraging object states *object-manipulating* and *context-manipulating* actions can be learnt. We propose a novel manipulation module, with object and context blocks, that capture the subtle similarities and differences between object states. We also show generalization to unseen objects and out-of-domain datasets. In future work, we intend to extend object states for procedure planning and action segmentation.

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

# A APPENDIX

## A.1 RELATED WORK

The study of object states and how they transform from one to the other has recently garnered interest in the computer vision community Isola et al. (2015); Yu & Grauman (2014); Nagarajan & Grauman (2018); Chen et al. (2020); Nan et al. (2019); Atzmon et al. (2020); Yang et al. (2020b); Misra et al. (2017); Wei et al. (2019); Bertasius & Torresani (2020); Liu et al. (2017); Fire & Zhu (2015); Zhou & Berg (2016). Understanding these states could enable an object recognition system to generalize better to unseen states, as studied in Bertasius & Torresani (2020); Misra et al. (2017). However, most studies have focused on image-based datasets Isola et al. (2015); Yu & Grauman (2014); Misra et al. (2017), where the goal is to capture the underlying distribution of states (*e.g*, different sizes of heels in shoes, old and new laptops). Some of these states are just *attributes* of objects (*e.g*, red car and blue car) and are generally static over time; whereas other states are transformations of objects over time (*e.g*, raw→ripe, whole→pieces, on table→on plate). These different types of states provide very different information. In this work, we leverage the state transformations that happen over time for understanding transforming actions. For the rest of this paper, we will refer to changes in object states (including visual appearance and context) over time as *object state transformations*.

For action recognition, most modern approaches can be grouped into two categories: (a) models utilizing temporal and spatial convolutions over videos to classify the actions Girdhar et al. (2017); Simonyan & Zisserman (2014); Feichtenhofer et al. (2016); Karpathy et al. (2014); Lin et al. (2019); Wang et al. (2016a); Yang et al. (2020a); Zhou et al. (2018a), and (b) models which rely on human-object interactions over time to recognize actions Gupta et al. (2009); Khan et al. (2012); Shen et al. (2018); Wu et al. (2019); Tan et al. (2019); Xiao et al. (2019). In both categories, the focus is generally to leverage as much temporal information as possible; this is evident with the focus on recent works to utilize as many frames as computationally feasible to understand actions.

Similar to our approach, Bakr et al. (2019); McCandless & Grauman (2013); Fathi & Rehg (2013) propose using state transformations for action recognition; *e.g*, Bakr et al. (2019) uses five key-frames for modeling transformations for action recognition. Closest to our approach is the work of Fathi and Rehg Fathi & Rehg (2013), which also uses two object bounding boxes as states for action recognition. However, our approach explicitly amplifies affinity and differential signals between states to extract *fluent* representations, which are used for action recognition.

## A.2 DATASET SPLITS EXTENSION

We use Epic-kitchens-55 (EK-55) Damen et al. (2018) as our primary dataset. EK-55 has kitchen activities captured with an egocentric camera. In total, it has 352 sparsely annotated bounding boxes for objects and 125 actions. Among all actions, only 13 actions are object-transforming and 30 actions are context-transforming. We choose these actions such that each action has atleast 40 videos in total. Our subset is also impacted by pre-processing step (object bounding boxes detection), since we require two bounding boxes of the object of interest either from the detector or the ground truth. This drastically reduces the number of videos with a long-tailed distribution over the classes. In total, we use 7342 videos, 2120 for object-transforming and 5222 for context-transforming actions.

Due to imbalanced number of object-manipulating and context-manipulating actions, we create five train/val/test splits of the dataset and report the mean performance across splits. Each split has all 13 object-transforming actions and randomly chosen 13 context-transforming actions. The test set in each split contains actions involving objects unseen during training. This is different from the typical test set provided for EK-55 Damen et al. (2018). We argue that using those splits do not provide evaluation of the model learning actions or latching onto object features. Therefore, by having different objects in train and test set inspired by Nagarajan & Grauman (2018), we ensure that the model learns the action, and does not rely only on object visual features. We refer to test set as "unseen object-actions" and validation set as "seen object-actions" set.

## A.3 PREPROCESSING: EXTRACTING OBJECT STATES

As discussed above, analyzing object state provides useful information for inferring actions. For this purpose, we first identify two frames in a video, one before and the other after the action has

Table 3: **Quantitative results on Seen Objects**. This table summarizes results for seen object actions (validation set). All values are mean with confidence interval (95%). In the table we highlight the best and second best result in bold+underline and underline respectively.

| Network Input | Method | All Actions | | Object Transforming Actions | | Context Transforming Actions | |
|---|---|---|---|---|---|---|---|
| | | Acc. | Mean Acc. | Acc. | Mean Acc. | Acc. | Mean Acc. |
| 8 Frames | TRN Zhou et al. (2018a) | $48.20 \pm 2.0$ | $11.68 \pm 1.6$ | $47.20 \pm 7.0$ | $14.18 \pm 1.9$ | $48.20 \pm 7.1$ | $9.07 \pm 2.1$ |
| | TPN Yang et al. (2020a) | $56.13 \pm 5.2$ | $14.11 \pm 1.8$ | $13.01 \pm 11.4$ | $18.42 \pm 3.2$ | $49.67 \pm 9.4$ | $9.60 \pm 3.2$ |
| 3 Frames | TRN Zhou et al. (2018a) | $45.68 \pm 5.4$ | $12.27 \pm 4.8$ | $44.16 \pm 8.3$ | $13.90 \pm 3.7$ | $47.50 \pm 8.3$ | $10.62 \pm 6.2$ |
| | TPN Yang et al. (2020a) | $35.70 \pm 2.0$ | $8.63 \pm 1.0$ | $32.64 \pm 7.0$ | $11.05 \pm 3.0$ | $39.52 \pm 10.1$ | $6.09 \pm 1.0$ |
| 8 Objects | TRN Zhou et al. (2018a) | $57.50 \pm 1.6$ | $17.90 \pm 2.9$ | $56.16 \pm 2.8$ | $20.77 \pm 2.5$ | $\underline{59.26 \pm 3.0}$ | $14.76 \pm 3.7$ |
| | TPN Yang et al. (2020a) | $59.92 \pm 2.9$ | $17.00 \pm 4.8$ | $65.28 \pm 4.6$ | $21.90 \pm 4.6$ | $50.84 \pm 5.6$ | $11.58 \pm 5.4$ |
| 3 Objects | TRN Zhou et al. (2018a) | $45.95 \pm 3.8$ | $13.62 \pm 3.8$ | $44.88 \pm 8.9$ | $15.36 \pm 5.3$ | $46.29 \pm 6.4$ | $11.86 \pm 5.5$ |
| | TPN Yang et al. (2020a) | $34.48 \pm 6.3$ | $8.57 \pm 3.5$ | $23.28 \pm 15.1$ | $8.97 \pm 6.1$ | $50.31 \pm 8.0$ | $8.38 \pm 1.6$ |
| 2 Objects | Fathi et al. Fathi & Rehg (2013) | $55.93 \pm 0.9$ | $15.34 \pm 3.1$ | $61.12 \pm 1.7$ | $20.15 \pm 4.8$ | $46.30 \pm 3.4$ | $10.26 \pm 1.6$ |
| | MaxpoolResFeats | $56.80 \pm 1.0$ | $18.47 \pm 8.2$ | $56.48 \pm 2.4$ | $21.35 \pm 9.9$ | $56.57 \pm 5.9$ | $14.40 \pm 7.0$ |
| | ConcatResFeats | $\underline{66.08 \pm 0.9}$ | $\underline{18.80 \pm 2.5}$ | $\underline{70.30 \pm 1.5}$ | $\underline{22.96 \pm 4.1}$ | $57.87 \pm 4.3$ | $\underline{15.26 \pm 2.5}$ |
| | **Ours** | $\mathbf{\underline{68.36 \pm 1.5}}$ | $\mathbf{\underline{20.50 \pm 2.8}}$ | $\mathbf{\underline{71.84 \pm 2.4}}$ | $\mathbf{\underline{24.91 \pm 2.1}}$ | $\mathbf{\underline{61.36 \pm 7.0}}$ | $\mathbf{\underline{16.00 \pm 3.9}}$ |

been performed, corresponding to different states of the object. We choose the highest confidence bounding box, extracted using an object detector, of the object of interest from each frame as the inputs to our model. We refer to these two regions as *start* and *end* states.

In particular, we leverage the temporal annotations provided with the video to identify clips where the action is performed. We extract bounding boxes using an object detector on the first $N$ frames of the first clip and the last $N$ frames from the last clip, and choose the box with highest class confidence score for the object under consideration as *start* and *end* states, respectively.

## A.4  IMPLEMENTATION DETAILS

We use ego-centric kitchen video dataset, Epic-Kitchen-55 (EK-55) Damen et al. (2018) for training. This dataset provides temporal action localization labels and has object-centric actions. Each video can have multiple clips of same object-action pair. We consider first and last clip for the same object-action in proximity, so that accurate *start* and *end* states are captured for objects. After merging the first and last clips from the video, we use one merged clip for each object-action pair from a video.

For extracting object states, use a Faster-RCNN Ren et al. (2015) detector, pre-trained on EK-55 Damen et al. (2018), which has sparsely annotated ground truth bounding boxes for 295 object classes. We apply it on two sets of images from an object-action clip with 30fps: first and last 20 frames. From each set, the object's bounding box with the highest confidence is chosen, which should be higher than 65%. This gives use *start* and *end* object states. Features are extracted using ResNet50 He et al. (2016) pretrained on ImageNet Deng et al. (2009) from the penultimate layer, and are of dimensions $1 \times 2048$. Extracted features are then passed through our *fluent* Attention module, and classified for all actions. For our model, we use learning rate 1e-5, Adam optimizer and ReLU activation. Since our dataset follows long-tail distribution, we use weighted cross-entropy loss over the actions. We empirically choose a temperature value as 100. We use L2-normalization on output of both context and object, before fusion.

## A.5  RESULTS AND ABLATION

We also show results on seen object in Table 3. Similar to unseen objects, our model outperforms previous models with a margin for all categories.

(***start→end***) **Attention:** We use a non-spatial attention block, inspired by Wang et al. (2018); He et al. (2020). Given an input feature (*start/end* state, denoted as $g_s/g_t$), three linear projections are applied to compute query ($q$) with respect for before state $g_s$, and key ($k$) and value ($v$) with respect to after state $g_t$. The attention ($a$) is weighted sum of value representation, where the weight ($M$) is

dot product between the query and the corresponding key, followed by a softmax normalization as

$$M = \text{softmax}\left(\frac{qk^T}{\lambda}\right)$$
$$a = Mv$$
$$o = g_s + \text{BatchNorm}(a)$$

where $o$ represents attention output for *start* state with respect to *end* state, and $\lambda$ is the temperature parameter. Note that $g_s, q, k, v, o, a \in \mathbb{R}^n; M \in \mathbb{R}^{n \times n}$, where $n = 2048$. This experiment shows we need different ways to amplify similarities and differences between *start* and *end* states.

