# OpenReview forum: "Recognizing Actions using Object States"
_ICLR.cc/2022/Workshop/OSC — ICLR2022 OSC  Poster_

### Official Review · Reviewer_TV5u · 2022-03-15
**comments for RECOGNIZING ACTIONS USING OBJECT STATES**

**Rating:** 2
**Confidence:** 3

**Review:**


The method proposed in this paper seems better than some other methods in the presented quantitative results. Moreover, it only uses two frames of image information to achieve these good results which is less costly. This method also shows strong generalization in unseen objects and unseen datasets. However, I think the design of your manipulation module, especially context module and object module, is not motivated well. Is this based on empirical results or other theoretical analysis (it is better to cite them if it exists)? Besides, there are some typographical errors such as Table ???. Check carefully again.

---

### Official Review · Reviewer_9gwa · 2022-03-17

**Rating:** 2
**Confidence:** 2

**Review:**

This paper is in the scope of the workshop and proposes to use the dot product similarity between object embeddings at two separate timesteps to infer the underlying action in a video.

There are some typos in the paper, for example missing table references on table 4. While it does seem like using the difference in object states to infer an action is good -- it feels like some actions are best inferred not from a change in object state, but rather the actions needed to move the object from one state to another. So, it may be interesting to see how to extend the work to a video of different frames.

Its interesting that the approach appears to outperform other baselines on this task -- why this happening? Is this generally applicable to other setting besides actions also?

---

### Decision · Program_Chairs · 2022-03-21

**Decision:**

Accept (Poster)

**Comment:**

The reviewers agree the paper should be accepted at the workshop. Congratulations!

The authors are encouraged to take the feedback by the reviewers into account and to fix the typographical errors when preparing the camera-ready version of the paper.